# Spatio-Temporal Dynamics of *Plasmodium falciparum* and *Plasmodium vivax* in French Guiana: 2005–2019

**DOI:** 10.3390/ijerph18031077

**Published:** 2021-01-26

**Authors:** Jenna Scully, Emilie Mosnier, Aurel Carbunar, Emmanuel Roux, Félix Djossou, Nicolas Garçeran, Lise Musset, Alice Sanna, Magalie Demar, Mathieu Nacher, Jean Gaudart

**Affiliations:** 1Department of Epidemiology, Mailman School of Public Health, Columbia University, New York, NY 10032, USA; 2Infectious and Tropical Disease Unit, Cayenne Hospital, 97306 Cayenne, French Guiana; emilie.mosnier@gmail.com (E.M.); felix.djossou@ch-cayenne.fr (F.D.); 3INSERM, IRD, SESSTIM, Health Economics & Social Sciences & Health Information Processing, Aix Marseille University, 13385 Marseille, France; 4Delocalized Prevention and Care Centers, Cayenne Hospital, 97306 Cayenne, French Guiana; aurel.carbunar@ch-cayenne.fr (A.C.); ngarceran@gmail.com (N.G.); 5ESPACE-DEV (IRD, University of Reunion Island, University of the West Indies, University of French Guiana, University of Montpellier), 34000 Montpellier, France; emmanuel.roux@ird.fr; 6LMI Sentinela, International Joint Laboratory ‘Sentinela’ (Fiocruz, UnB, IRD), Rio de Janeiro, RJ 21040-900, Brazil; 7Amazonian Ecosystems and Tropical Diseases, EA3593, University of French Guiana, 97300 Cayenne, French Guiana; 8Parasitology Laboratory, Malaria National Reference Center, French Guiana Pasteur Institute, 97300 Cayenne, French Guiana; lmusset@pasteur-cayenne.fr; 9French Guiana Regional Health Agency, 97306 Cayenne, French Guiana; alice.sanna@ars.sante.fr; 10Parasitology & Mycology Laboratory, Cayenne Hospital, 97306 Cayenne, French Guiana; magalie.demar@ch-cayenne.fr; 11French Guiana and West Indies Clinical Investigation Center-INSERM 1424, Cayenne Hospital, 97306 Cayenne, French Guiana; mathieu.nacher66@gmail.com; 12Aix Marseille University, IRD, INSERM, APHM, La Timone Hospital, Biostatistics and ICT, 13385 Marseille, France

**Keywords:** malaria, meteorological factors, *Plasmodium vivax*, *Plasmodium falciparum*, Amazonia, French Guiana, hotspots

## Abstract

*Aims:* This study examines the dynamics of malaria as influenced by meteorological factors in French Guiana from 2005 to 2019. It explores spatial hotspots of malaria transmission and aims to determine the factors associated with variation of hotspots with time. *Methods*: Data for individual malaria cases came from the surveillance system of the Delocalized Centers for Prevention and Care (CDPS) (*n* = 17) from 2005–2019. Meteorological data was acquired from the NASA Goddard Earth Sciences Data and Information Services Center (GES DISC) database. The Box–Jenkins autoregressive integrated moving average (ARIMA) model tested stationarity of the time series, and the impact of meteorological indices (issued from principal component analysis—PCA) on malaria incidence was determined with a general additive model. Hotspot characterization was performed using spatial scan statistics. *Results*: The current sample includes 7050 eligible *Plasmodium vivax* (*n* = 4111) and *Plasmodium falciparum* (*n* = 2939) cases from health centers across French Guiana. The first and second PCA-derived meteorological components (maximum/minimum temperature/minimum humidity and maximum humidity, respectively) were significantly negatively correlated with total malaria incidence with a lag of one week and 10 days, respectively. Overall malaria incidence decreased across the time series until 2017 when incidence began to trend upwards. Hotspot characterization revealed a few health centers that exhibited spatial stability across the entire time series: Saint Georges de l’Oyapock and Antecume Pata for *P. falciparum*, and Saint Georges de l’Oyapock, Antecume Pata, Régina and Camopi for *P. vivax. Conclusions*: This study highlighted changing malaria incidence in French Guiana and the influences of meteorological factors on transmission. Many health centers showed spatial stability in transmission, albeit not temporal. Knowledge of the areas of high transmission as well as how and why transmission has changed over time can inform strategies to reduce the transmission of malaria in French Guiana. Hotspots should be further investigated to understand other influences on local transmission, which will help to facilitate elimination.

## 1. Introduction

Understanding the dynamics of malaria transmission is crucial for the development and implementation of successful interventions that could reduce risk. Malaria transmission occurs in 91 countries and accounted for 405,000 deaths in 2018 [1]. French Guiana (83,534 km^2^ for 269,352 inhabitants in 2016, [2]) is a French overseas department located in the Amazonian region of South America. About 90% of French Guiana is covered by rainforest, and deforestation leads to dense vector populations [2,3,4,5]. Risk of malaria transmission is high in gold-mining regions, generally in the geographical center of the department, as well as among Amerindians, along sections of the borders with Brazil and Suriname [4,6]. Much of the endemic region is occupied by remote villages, and those at greatest risk of malaria are the estimated 6500 people living in these villages, and the 5000 to 10,000 garimpeiros, or illegal Brazilian gold miners, in the forest [4,5]. In addition to gold miners and indigenous persons, a high prevalence of cases is seen among immigrants, the French army, and travelers who do not follow prevention protocol [4,7]. The predominant mosquito vector in French Guiana is *Anopheles darlingii* [3].

It has been previously shown that malaria transmission follows seasonal trends; in French Guiana there is a short rainy season from mid-December to February, a dry season in March, a rainy season from April to mid-July, and a dry season from mid-July to mid-December [8,9]. Malaria incidence is strongly influenced by environmental factors that determine the availability and productivity of *Anopheles* habitats. One previous study described a model based on meteorological data and satellite imagery of the landscape to predict vector densities in coastal areas of Cayenne and Cacao village, and at the border between French Guiana and Brazil [10,11,12]. There is no current literature regarding malaria incidence in neighboring Suriname, or on the impact of the relationship between deforestation, meteorological factors, and malaria cases in Brazil and French Guiana [4,13,14]. No previous research in French Guiana has covered the entire department, but rather focused on specific regions or used surveillance data that covered such a long time period. Examining the hotspots of malaria transmission, particularly in times of low transmission, can help to reveal environmental and behavioral factors of transmission and targets for intervention in endemic regions [15,16,17].

We used data from an established surveillance system to record malaria cases and determine the association between meteorological factors and transmission [18,19]. This study examines the dynamics of malaria as influenced by meteorological factors in French Guiana from 2005 to 2019. It explores spatial hotspots of malaria transmission and aims to determine the factors associated with the variation of hotspots from period to period.

## 2. Materials and Methods

### 2.1. Data Source

Anonymized data from individual malaria cases were obtained monthly from the Delocalized Centers for Prevention and Care (CDPS) surveillance system operated by the Cayenne Hospital, which has been in operation since 2005. There are 17 CDPS health centers present in French Guiana that deliver malaria diagnosis and treatment (Figure 1). A malaria case is defined as any positive malaria Rapid Diagnosis Test (RDT, SD Bioline^®^ Malaria Ag Pf/Pan test; pfHRP2/pLDH-based Standard Diagnostic, Inc., Gyeonggi, Republic of Korea) for *Plasmodium falciparum* or *Plasmodium vivax* performed and recorded in the CDPS system. This database is representative of the rural areas of French Guiana from September 2005 until April 2019.

### 2.2. Study Area

Global positioning system (GPS) coordinates for each health center were determined from an open street map by the regional health agency of French Guiana [21]. The estimated population was based on census data published by the Institut National de la Statistique et des Études Économiques (INSEE) for 2006, 2010, and 2016, and projections were estimated for gaps in these years [2].

### 2.3. Case Definition

New attacks of malaria in this database are unable to be distinguished from malaria notifications related to the follow-up of the patients, treatment failures, or *P. vivax* relapses. For this reason, we followed previous research regarding the classification differentiation and considered that a *P. vivax* malaria notification could be considered a new case if it occurred more than 90 days after the last positive result, and a notification occurring within 90 days was considered a relapse [5]. In order to account for *P. falciparum* treatment failure and follow-up visits, notifications within seven days of previous contamination were considered the same infection (Figure 2).

### 2.4. Meteorological Data

Monthly meteorological data were obtained from NASA Goddard Earth Sciences Data and Information Services Center (GES DISC) for the same study period [18,19]. The meteorological variables included were rainfall (mm), surface air temperature (°C, minimum and maximum), and specific humidity (g/kg, minimum and maximum). All meteorological data were measured at 0.1° × 0.1° resolution. The meteorological factors were analyzed with principal component analysis to reduce collinearities and dimensions.

### 2.5. Statistical Analyses

A change-point analysis was conducted to detect high and low malaria transmission periods. The change-point analysis in variance was performed using the pruned exact linear time algorithm (PELT). The transmission periods determined via change-point analysis were mapped, and hotspots were identified using Kulldorff’s spatial scan statistic [22]. The Discrete Poisson model algorithm used circular spatial windows that were centered around each health center. The maximum potential spatial cluster size was 50% of the population at risk. Spatial cluster analysis was performed to determine the dynamics of transmission in the study area. The incidence for *P. falciparum* and *P. vivax* was mapped by health center.

The Box–Jenkins autoregressive integrated moving average (ARIMA) procedure was used to achieve stationarity, and cross-correlation functions were used to assess lag between the combined meteorological factors and malaria incidence. Generalized additive models (GAM) were designed to assess the impact of each meteorological component and time on malaria incidence.

The statistical analyses were performed with R software version 3.5.2 (R Development Core Team, R Foundation for Statistical Computing, Vienne, Austria). Local hotspot assessment was performed by using the SatScan™ software version 9.4.2. (SatScan, Boston, MA, USA).

### 2.6. Legal and Ethical Considerations

The CDPS database was anonymized and declared to the Commission Nationale Informatique et Libertés (CNIL) (authorization N°1939018).

## 3. Results

### 3.1. Study Population

There were 7050 total cases of malaria included in the analysis for the period between September 2005 and April 2019: 2939 *P. falciparum* and 4111 *P. vivax*. Of the individuals with *P. falciparum,* the average age at diagnosis was 21.14 ± 17.13 and 21.10% of those were ≤5 years of age. Of those with *P. vivax,* the average age at diagnosis was 20.68 ± 16.95 and 18.54% of those were ≤5 years of age, indicating no significant difference between *P. falciparum* and *P. vivax* in terms of age at diagnosis (*p* = 0.27). Males accounted for 54.8% of *P. falciparum* cases, and females accounted for 45.2%. Of the *P. vivax* cases, 52.4% were males, and 47.6% were females.

### 3.2. Overview of the Time Series

The malaria incidence plot of the time series in Figure 3 shows an association between malaria cases and dry/rainy periodicity as well as temperature and humidity patterns. The highest incidences of both *P. falciparum* and *P. vivax* fell between October and January. These peaks in incidence coincided with peaks in maximum temperature and dips in minimum humidity. We also observed different peaks according to the *Plasmodium* species, particularly between 2005 and 2009.

### 3.3. Malaria Transmission Periods

Change-point analysis identified four distinct periods of malaria transmission across the time series. There was a high transmission period from September 2005 until January 2009 where the average incidence rate of malaria was 4.63 per 10,000 people. Following this period was a moderate one from February 2009 until March 2013 where the average incidence rate was 1.76 cases per 10,000 people. A low transmission period lasted from April 2013 until August 2017 where the average incidence rate was 0.38 cases per 10,000 people. Finally, there was a slight peak in transmission from September 2017 until April 2019 where the average incidence rate was 0.85 cases per 10,000 people.

Within the highest transmission period from September 2005 to January 2009, 2082 (51.4%) cases were *P. vivax*. From February 2009 to March 2013, 1202 (59.4%) cases were *P. vivax*, and from April 2013 to August 2017, 406 (79.1%) cases were from *P. vivax*.

Table 1 shows malaria incidence disaggregated by health center by each transmission period. The health centers with the highest incidence of cases were Antecume Pata, Camopi, Ouanary, and Saül. In Cacao, Ouanary, and Saint Georges de l’Oyapock, which are all located in the northeast of the department, the incidence of *P. vivax* was higher than the incidence of *P. falciparum* in the highest transmission period. We also saw this trend in Trois Sauts. The most notable spikes in incidence late in the time series were at Régina, Saint Georges de l’Oyapock, and Taluen.

Figure 4 shows the spatial hotspot distribution for both *P. falciparum* and *P. vivax* across each transmission period. Hotspots for *P. falciparum* (Figure 4A) relative to other locations during each period were consistent at Antecume Pata and Saint Georges de l’Oyapock. The highest-risk hotspot in the first transmission period had a risk ratio (RR) of 73.33 (*p* < 0.0001); the health center was in rural Camopi with a population of 1514 people. During the second transmission period, the highest-risk cluster (RR = 139.00, *p* < 0.0001) was in Saül where inhabitants visited the health center from both urban and rural neighborhoods. In the third transmission period, the highest-risk cluster (RR = 30.02, *p* < 0.0001) was Trois Sauts, and during the fourth period, Saint Georges was the highest-risk cluster (RR = 31.94, *p* < 0.0001).

*Plasmodium vivax* hotspots (Figure 4B) were constant at Antecume Pata, Régina, Saint Georges de l’Oyapock, and Camopi. The highest-risk *P. vivax* cluster during the first transmission period was at Camopi (RR = 41.91, *p* < 0.0001), at Saül (RR = 73.85, *p* < 0.0001) during the second period, and at Saint Georges de l’Oyapock during the third (RR = 49.02, *p* < 0.0001) and fourth transmission periods (RR = 91.17, *p* < 0.0001).

### 3.4. Meteorological Data and Malaria Incidence Analysis

It is difficult to assess the individual crude effect of each meteorological factor; therefore, principal component analysis was used to analyze the combination of variables associated with malaria. Principal component analysis identified three main meteorological components that represented 97.8% of the total variance. The first component (72.64% of the variance) was made up of maximum and minimum temperatures and minimum humidity. The second component (17.2% of the variance) was composed of maximum humidity, and the third component (8% of the variance) was constituted by rainfall.

The first meteorological component (maximum and minimum temperature and minimum humidity) was significantly negatively correlated with total malaria incidence with a lag of one week (correlation coefficient: −0.239). The second component (maximum humidity) exhibited a significant negative correlation with total malaria incidence (correlation coefficient: −0.251) and a lag of about 10 days. After considering the relationship between temperature, humidity, and malaria, there was a decreasing trend in malaria incidence as temperature and humidity increase. Finally, the third component (rainfall) was not significantly correlated with malaria incidence.

Figure 5 shows the generalized additive models (GAM) for the association between each meteorological component (explaining 97.8% variance) and time with malaria incidence. Malaria cases were aggregated for both parasite species for this modelling procedure. The first meteorological component (Figure 5a, maximum and minimum temperature and minimum humidity) shows a negative linear downward trend in malaria incidence as temperature and minimum humidity increased. The maximum humidity component (Figure 5b) shows a quasi-linear relationship, indicating that malaria incidence declined slowly until a threshold where incidence decreased more rapidly as humidity increases. A negative trend was observed for time (Figure 5c), indicating decreasing malaria incidence in French Guiana until 2017 when there was an increase in cases.

## 4. Discussion

This study examined the dynamics of malaria as influenced by meteorological factors in French Guiana from 2005 to 2019. It explored changes in incidence and spatial hotspots of malaria transmission. During the time series, malaria incidence decreased significantly across the department, which could be due to new malaria control tools and treatment interventions such as free malaria treatment for all, distribution of insecticide-treated nets, indoor residual spraying, Palustop and Malakit [6,23,24,25,26]. Malakit is a pilot project targeting self-diagnosis and self-treatment of malaria among illegal gold miners in the Guiana Shield, and Palustop is a study evaluating the effect of treating both symptomatic and asymptomatic *Plasmodium* infections [6,26]. The transmission periods determined by change-point analysis followed patterns of malaria intervention implementation. In 2002, “Anaconda and Toucan” was implemented by the French army to reduce illegal gold mining, followed by “Operation Harpie” in 2008, as well as the introduction of artemisinin-based therapies in CDPS health centers [25]. These interventions were implemented at the end of the first transmission period (September 2005 to January 2009), and likely influenced the reduction in cases that marked the beginning of the second transmission period. The lowest transmission period (April 2013 to August 2017) coincided with the use of artemisinin therapies, the fight against illegal gold mining, and campaigns for free distribution of long-lasting insecticide-treated nets. The latest peak in transmission from September 2017 to April 2019 could be the result of resurgence from transborder mobility or deforestation [27,28].

Previous research has found that malaria transmission is driven by environmental factors and understanding these factors can inform where and when to reinforce control strategies [29]. Rainfall has been previously linked with malaria transmission, as rain can create and destroy mosquito breeding sites, however this study showed no significant correlation between rainfall and malaria incidence [30,31]. Furthermore, the component made up of maximum and minimum temperature and minimum humidity was negatively correlated with malaria incidence with a lag of one week. As temperature and humidity increased, a decrease in malaria incidence was seen one week later. Increasing temperatures have been previously found to increase malaria incidence up to a threshold, however, these results showed no positive relationship between the component for temperature and minimum humidity with malaria [32,33,34,35].

A few of the health centers were identified as hotspots via SaTScan across the entire time series, indicating spatial stability: Saint Georges de l’Oyapock and Antecume Pata for *P. falciparum*, and Saint Georges de l’Oyapock, Antecume Pata, Régina, and Camopi for *P. vivax*. The health center at Taluen exhibited a spike in *P. vivax* cases towards the end of the time series after reporting no malaria cases in the highest transmission period. Antecume Pata, Camopi, and Saül reported higher cases counts for *P. falciparum* during the highest-risk malaria periods, whereas Cacao, Ouanary, and Saint Georges de l’Oyapock had higher *P. vivax* transmission. The health centers in Ouanary and Saint Georges de l’Oyapock are about 38 km away from each other, and about 97 km from Ouanary to Cacao, which could indicate a pocket of *P. vivax* concentrated in this region. Additionally, the areas with highest *P. vivax* incidence were Ouanary, Saint Georges de l’Oyapock, Camopi, Trois Sauts, Saül, and Antecume Pata, all of which are close to the Brazil border except Antecume Pata and Saül. Antecume Pata is located along the border with Suriname, and mining around Saül attracts people from throughout the Guiana Shield [4,7,36]. We observed an endemo-epidemic stable border area due to remoteness that allowed for reviviscence, limited access to health care for gold miners and indigenous persons, and specific risk factors within these communities including fishing, hunting, and gold mining [4,7]. Malaria transmission was consistently low along the coast and in the northwest of the department, where most of the population lives.

There are multiple limitations to this study in terms of malaria case counts and statistical methods. First, there is often underestimation of cases in endemic areas, and additionally in regions with mobile populations [4,7]. Individuals may seek malaria treatment in Suriname or Brazil, or reciprocally, which could under or overestimate the prevalence of malaria in French Guiana [23]. Secondly, there are high levels of asymptomatic malaria carriage, which is characteristic of endemic areas, so it is likely that infected individuals are not being picked up by the CDPS system [6,36]. Third, French Guiana does not have a dense population, and many of the health centers are located far from each other. This separation could limit the SaTScan Discrete Poisson model algorithm, which used circular spatial windows that were centered around each health center. Finally, this paper did not examine other drivers of malaria transmission including human behavior, “cross-border malaria,” antimalarial or insecticide resistance, malaria self-treatment, the ecological impacts of illegal gold mining, or the influence of control interventions, all of which greatly influence transmission dynamics.

## 5. Conclusions

This study highlighted changing malaria incidence in French Guiana and the influences of meteorological factors on transmission. Many health centers showed spatial stability in transmission, albeit not temporal stability. Further research should be performed in French Guiana to understand the relationships between meteorological factors, ecological impacts, and human behaviors on malaria transmission. Knowledge of the areas of high transmission as well as how and why transmission has changed over time can inform strategies to reduce the transmission of malaria in French Guiana.

## Figures and Tables

**Figure 1 ijerph-18-01077-f001:**
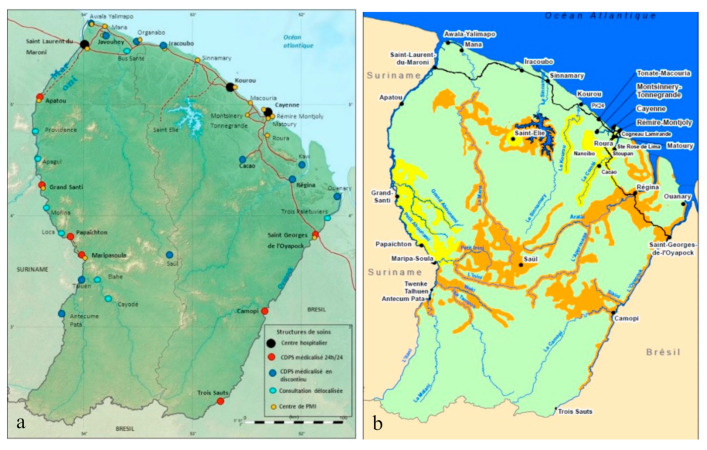
(**a**) Locations of Delocalized Centers for Prevention and Care health centers and hospitals in French Guiana. Malaria risk map (**b**), green areas are where there was no malaria transmission in the previous year (2017), yellow areas indicate low risk of malaria but no transmission in the villages, and orange areas indicate high risk of malaria transmission [20]. Figure 1b has been produced with the assistance of: French Guiana territorial collective, French Guiana Health Agency, Malaria National Reference Center of French Guiana, French Guiana armed forces, Delocalized Prevention and Care Centers (CDPS) of Cayenne Hospital, Medical Biology Laboratory of French Guiana, Cire of French Guiana and France Public Health Institution.

**Figure 2 ijerph-18-01077-f002:**
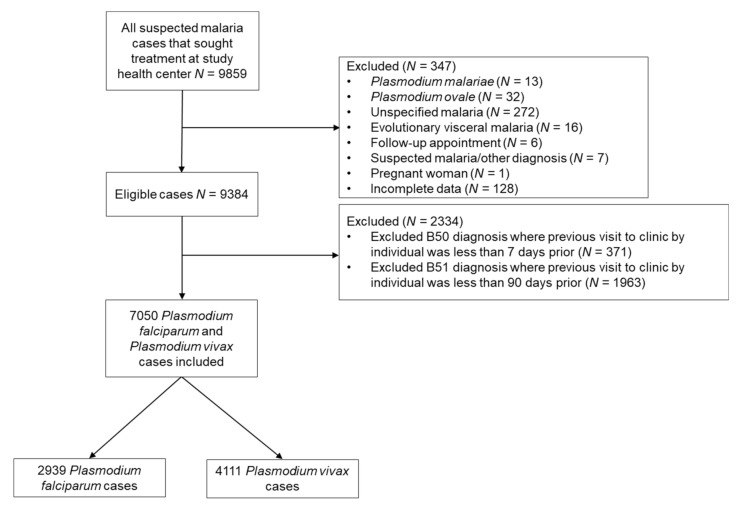
Participants included in the analysis. Codes B50 and B51 refer to the CIM10 codification for *P. falciparum* and *P. vivax*, respectively.

**Figure 3 ijerph-18-01077-f003:**
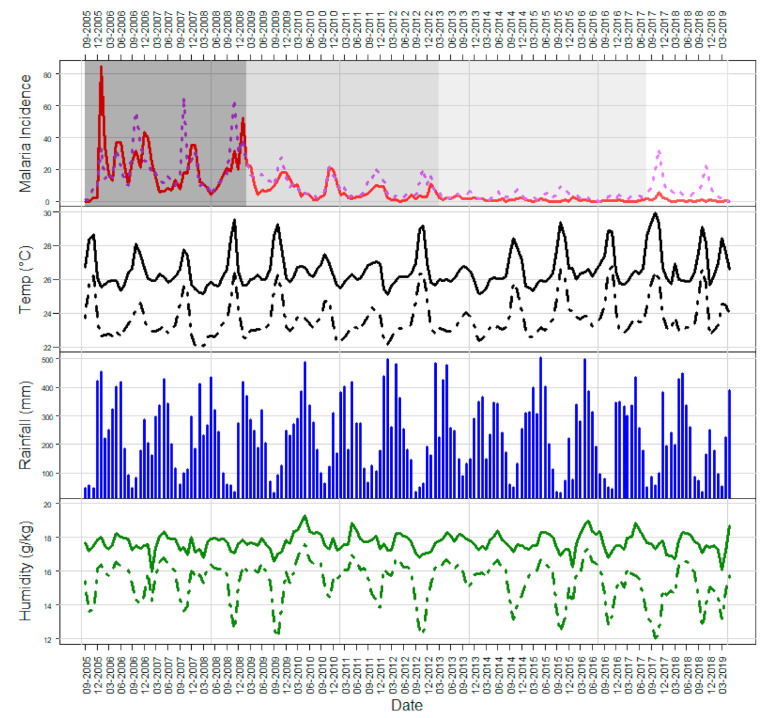
Monthly meteorological factors and malaria incidence from 2005 to 2019. The first plot shows malaria incidence (per 100,000 people, red curve *P. falciparum*, purple dotted curve *P. vivax*), the second plot represents maximum and minimum temperature (°C, respectively solid and dashed black curves), the third plot shows rainfall (mm), and the last plot represents maximum and minimum humidity (g/kg, respectively solid and dashed green curves). The white/grey background in the first plot represents the different transmission periods identified by change-point analysis.

**Figure 4 ijerph-18-01077-f004:**
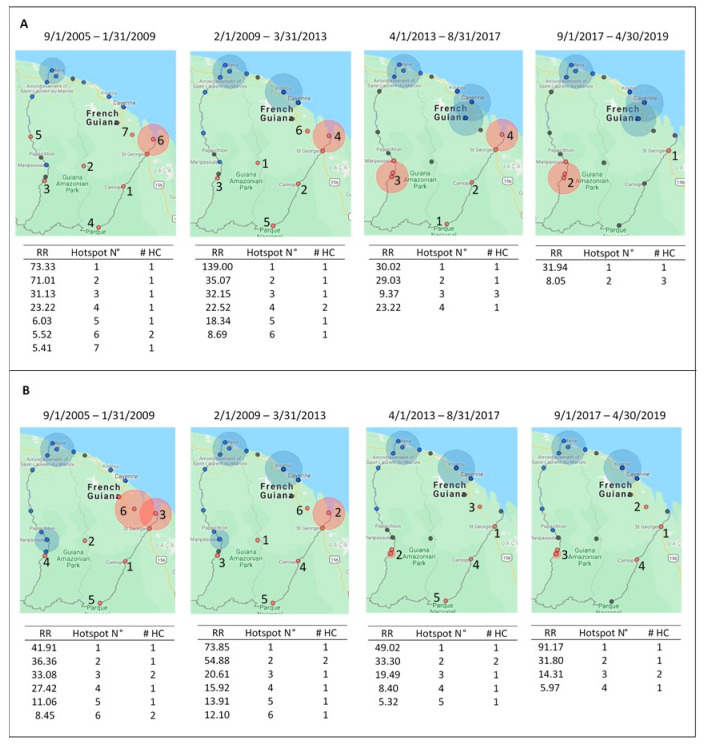
The red circles represent the high-risk clusters for *P. falciparum* (**A**) and *P. vivax* (**B**) during each transmission period. The blue circles indicate cold spots, and the black circles indicate neutral spots relative to the other locations. The attached tables present the risk ratios (RRs) for each hotspot along with the hotspot number (N°) corresponding to the number on the map, and the number of health centers within each cluster (# HC).

**Figure 5 ijerph-18-01077-f005:**
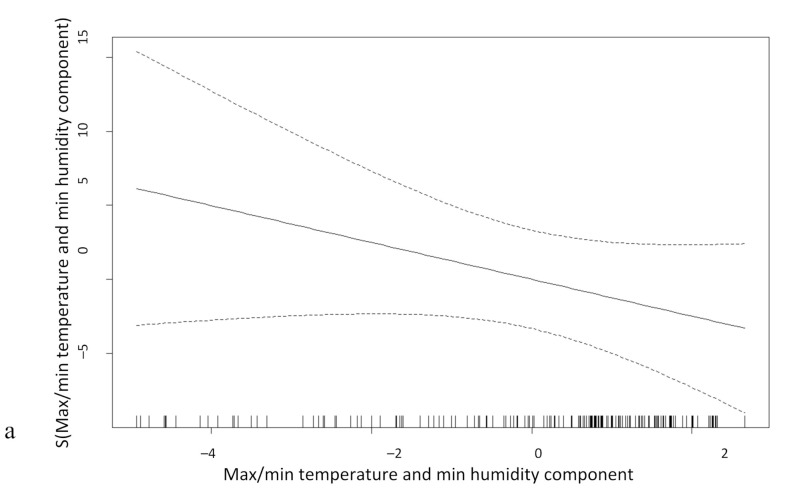
Generalized additive model for relationship between total malaria incidence (*P. falciparum* and *P. vivax*) and the first meteorological component (**a**—maximum and minimum temperature and maximum humidity), the second component (**b**—maximum humidity), and time (**c**). The solid black line indicates the smooth relationship with 95% confidence intervals (dashed lines).

**Table 1 ijerph-18-01077-t001:** Incidence per 1000 people for *Plasmodium falciparum (Pf)* and *Plasmodium vivax* (*Pv*)at each health center during each change-point analysis transmission period.

Health Center	September 2005–January 2009		February 2009–March 2013		April 2013–August 2017		September 2017–April 2019	
	*Pf* per 1000	*Pv* per 1000	*Pf* per 1000	*Pv* per 1000	*Pf* per 1000	*Pv* per 1000	*Pf* per 1000	*Pv* per 1000
Apatou	5.44	0.18	0.29	0.29	0.2	0	0	0
Antecume Pata	403	372.9	153.1	146.6	2.5	40.0	2.2	26.1
Awala Yalimapo	0	1.6	0	0.8	0	0	0	0
Camopi	574	420.1	131.9	102.1	13.4	17.4	1.6	12.3
Cacao	12.3	149.7	9.7	22.4	0	3.0	0	1.2
Gran Santi	71.8	2.7	0.4	0.7	0.3	0.4	0	0.1
Iracoubo	0	0	0	1.0	0	0	0	0.6
Javouney	3.6	0.9	0	0	0	0.7	0	0
Maripa-Soula	5.5	2.4	2.8	2.3	3.5	3.8	1.0	1.8
Organabo	4.4	0	0	0	0	0	0	0
Ouanary	46.2	246.2	82.6	128.4	16.5	0	0	0
Papaїchton	12.2	2.6	5.8	2.9	0.6	1.7	0.1	0.1
Régina	71.4	70.2	42.0	84.0	1.1	39.5	1.1	60.2
Saül	897	496.8	620.9	509.8	13.3	6.6	0	20.0
Saint Georges de l’Oyapock	67.3	256.0	73.9	175.4	2.9	52.5	3.9	67.6
Trois Sauts	28	150.7	86.6	97.6	15.9	11.6	2.8	7.0
Taluen	0	0	0	15.6	0	108.0	0	34.8

## Data Availability

The datasets generated and analyzed during the current study are not publicly available due to special authorization to transfer databases given by the Commission Nationale Informatique et Libertés (CNIL). Upon prior authorization by the CNIL, the dataset would be available from corresponding author on reasonable request.

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
