# Peer review of "Spatio-Temporal Dynamics of Plasmodium falciparum and Plasmodium vivax in French Guiana: 2005–2019"

_ijerph, 2021, doi:10.3390/ijerph18031077_

Round 1

Reviewer 1 Report

This paper uses a long-term dataset of malaria case incidence in rural areas of French Guiana (FG) to assess changes in case numbers over time and to identify temporal and spatial hotspots. Despite the limitations of the data as outlined in the Discussion by the authors, this paper provides a comprehensive view of occurrence of malaria in FG and uses appropriate statistical analyses to determine temporal and spatial patterns in the data. Congratulations to the authors on a well-written paper. Minor comments are as follows:

Abstract:

Line 39:

“(maximum/minimum temperatures/minimum humidity and maximum humidity, respectively)” change to “(maximum/minimum temperatures and maximum/minimum humidity, respectively)”.

Line 46: Please be more specific and detail the correlations with meteorological factors here.

Introduction:

Can you add some more background information here? Can you expand on previous malaria research in FG? What meteorological factors have an influence on malaria incidence in nearby countries such as Brazil? Is there any prior relevant research in Suriname?

Material and Methods:

Line 101: How did you estimate the gaps? Do you have a plot showing the population trend?

2.3 Case definition: can you explain/cite literature as to why you chosen 90 day cut off for relapse in vivax and 7 day cut for recrudescence in falciparum. Evidence is needed to back up these thresholds.

2.4 Met Data: What is resolution of the data, how is temp measured (distance above ground?) and are these daily min/max or monthly? Is rainfall mm per year or month?

Results:

3.2 Reference Figure 3 in the text.

Figure 5: Temperature is spelled wrong in a)

Discussion:

There is a clear positive correlation between increased temperature and malaria incidence shown in Figure 3 (including a visible lag phase), and yet this is not reflected in the PCA – why?

Reviewer 2 Report

Dear authors. 

I have read your manuscript and its nicely written. 

General comments: Results were written in such a way that non-experts will have hard time understanding implications of findings from this study. Analysis of PCA will not be useful to policy makers if not written in plan English, explaining the findings and their implications.

Major observations I have are formatting issues and typing errors. Also in the Results if there is a way to improve Figure 5 that will be good. 

For the PCA it will be better to create a figure showing for all observations and localities how variables cluster in space (showing for example how temperature and humidity cluster together with respect to malaria incidence, away from rainfall). That will be more informative than Figure 5. 

Please find my observations in the attached word document. 
